# The Microbiome as a Therapeutic Target for Multiple Sclerosis: Can Genetically Engineered Probiotics Treat the Disease?

**DOI:** 10.3390/diseases8030033

**Published:** 2020-08-30

**Authors:** Hannah M. Kohl, Andrea R. Castillo, Javier Ochoa-Repáraz

**Affiliations:** Department of Biology, Eastern Washington University, Cheney, WA 99004, USA; hkohl@eagles.ewu.edu (H.M.K.); acastillo@ewu.edu (A.R.C.)

**Keywords:** probiotics, microbiota, microbiome, multiple sclerosis, EAE

## Abstract

There is an increasing interest in the intestinal microbiota as a critical regulator of the development and function of the immune, nervous, and endocrine systems. Experimental work in animal models has provided the foundation for clinical studies to investigate associations between microbiota composition and function and human disease, including multiple sclerosis (MS). Initial work done using an animal model of brain inflammation, experimental autoimmune encephalomyelitis (EAE), suggests the existence of a microbiota–gut–brain axis connection in the context of MS, and microbiome sequence analyses reveal increases and decreases of microbial taxa in MS intestines. In this review, we discuss the impact of the intestinal microbiota on the immune system and the role of the microbiome–gut–brain axis in the neuroinflammatory disease MS. We also discuss experimental evidence supporting the hypothesis that modulating the intestinal microbiota through genetically modified probiotics may provide immunomodulatory and protective effects as a novel therapeutic approach to treat this devastating disease.

## 1. Introduction

Multiple sclerosis (MS) is a debilitating autoimmune disease of the central nervous system (CNS) that constitutes a devastating medical and economic burden for patients. In MS, immune cells enter the CNS and mistakenly attack and degrade the myelin sheath, which forms the protective covering for neurons, causing the formation of sclerosed plaques from which the disease gets its name [1]. MS lesions are observed by magnetic resonance imaging (MRI) [2], and although natural remyelination can occur, the process becomes less efficient as the disease progresses [3]. As the disease develops, axonal damage occurs causing a broad array of symptoms that include muscle weakness, loss of coordination, numbness, and double or blurring vision. At later stages MS patients may suffer from severe paralysis and pain, depression, and disturbances in urinary, sexual, and gastrointestinal functions [4]. Relapsing-remitting MS (RRMS) is the most common form of the disease [5]. Approximately, half of RRMS patients develop secondary progressive MS (SPMS) as a later stage of the disease, characterized by a gradual neurologic decline and reduced numbers of remissions and relapses [5]. The demyelination observed during MS may be the result of proinflammatory responses triggered by CNS-resident immune cells and by peripheral cells that cross the blood–brain barrier (BBB) and infiltrate into the parenchyma. T cells are the primary immunopathogenic cells in MS [6]. Both CD8^+^ and CD4^+^ T cells release proinflammatory cytokines that recruit additional T cells. Interleukin-17 (IL-17) and granulocyte-macrophage colony-stimulating factor (GM-CSF) are two of the most prominent proinflammatory cytokines associated with MS disease pathology [7]. Studies in mice lacking these cytokines are resistant to the induction of MS model disease [8]. IL-17 and GM-CSF promote the recruitment of additional immune cells into the CNS, leading to myelin sheath degradation and damage of the neuronal axons [9].

Despite efforts in the recent decades, the etiology of MS remains poorly understood. From 1990 to 2015, there has been a 59% increase in the global incidence of MS [10]. Both genetic and environmental factors have been linked to the onset of MS. Females are affected at a rate three times that of males [11]. Inherited genetic susceptibility is a contributing risk factor for MS. Studies examining MS incidence among twins indicate greater disease concordance rates among dizygotic twins (~30%) as compared to monozygotic twins (~5%) [12]. Ethnicity also appears to contribute to disease severity, as discussed later in this review [13,14,15]. Several genetic polymorphisms have been identified in human leukocyte antigen (HLA) genes [16] associated with increased risk for the development of MS [12]. A recent genome-wide association study (GWAS) of MS and healthy individuals revealed more than 200 GWAS associations independent of the major histocompatibility complex (MHC) molecules encoded by HLA genes in T cells, B cells, and monocytes, all of which play an important role in the immunopathology of the disease [17]. Additional polymorphisms have been identified in immune system genes, including the interleukin 7 receptor (*IL7R*) [18] and the vitamin D receptor (*VDR*) [19].

Environmental factors, including diet, exercise, pathogens, microbiota, vitamin D levels, and stress, are also associated with MS [20]. Many of these factors are known to affect the intestinal microbiota (also referred to as gut microbiota). Among the environmental factors linked to MS, the microbiome has attracted extensive interest over the last decade. The term microbiome is used widely in the literature, particularly in the most recent years. As reviewed by Berg and colleagues, the definition of microbiome might require standardization [21]. One of the most commonly cited definitions for the microbiome was provided by Lederberg and McCray, naming it as the “community of commensal, symbiotic, and pathogenic microorganisms within a body space or other environment” [22]. While the microbiota is the combination of all microorganisms living in a particular environment or habitat, the microbiome is summarized as the complete habitat, microbes, and “theatre of activity [23]”, including genomes, environmental factors, metabolic activity, and ecological function [21]. Despite the overwhelming interest in the biological and clinical repercussions of microbiome alterations, the complexity of the microbial community and multifactorial and bidirectional interactions with the host present significant technical challenges for this area of research, and additional tools and research are needed for a more complete understanding of how the host microbiome impacts disease. Over the last decades, the Food & Drugs Administration (FDA) and other worldwide agencies approved several compounds with anti-inflammatory effects for MS treatment. Most approved disease-modifying therapies (DMTs) are being used to treat RRMS. In general, DMTs target neuroinflammation but are limiting in terms of efficacy and affordability and are associated with a wide range of adverse effects (Table 1). In this review, we discuss how the host microbiome influences neuroinflammation and immunity and theorize that genetically engineered probiotics designed to enhance or provide protection against neuroinflammation might be used as immunomodulatory therapeutics for the treatment of MS.

## 2. The Crosstalk between the Intestinal Microbiome and the Host

### 2.1. The Intestinal Microbiome

The microbiota refers to the microorganisms within a defined environment while the microbiome refers to the entire habitat, including their genomes, the host, and the environmental conditions that define them [24]. The human intestinal microbiota is composed of up to 100 trillion microorganisms. Of the vast diversity of the microbial world so far identified, only a small percentage are represented in the human intestinal microbiota. The two main phyla are Bacteroidetes and Firmicutes [25], followed by Proteobacteria and Actinobacteria [26]. In addition, there is significant intestinal microbiome variability between healthy individuals, as well as within an individual over their lifespan. These differences may reflect host genetics and environmental factors, including diet, living conditions, and smoking, among other variables. In a study of 37 individuals whose microbiomes were sampled regularly, it was found that of the 200 bacterial strains compared, 60% were unchanged within a period of five years. Yet, when diet was modified to a liquid-based low-caloric diet, the subjects’ microbiome changed markedly [27], revealing that diet is a major contributor to intestinal microbial composition. Furthermore, these diet-dependent changes in microbiota composition were observed within a few days [28,29]. Additionally, age [30], sex [31], host immune system and genetics [32], infectious disease [33], and antibiotic use [34] can all impact intestinal microbiome composition in a given individual [35]. Therefore, although the human microbiota can be remarkably stable over time, studies have shown it is also malleable, suggesting it could be a target for therapeutic manipulation.

### 2.2. Impact of the Intestinal Microbiome on the Immune System

The role of the intestinal microbiota in regulating the immune system and inflammation levels has been extensively explored over the last several decades. Intestinal microbes are required for the proper development of a balanced immune system [36], and are believed to train the immune system in both an effective immune response against pathogens and immunological tolerance. This is evidenced in studies using germ-free (GF) mice lacking intestinal microbiota-derived signals [32], whereby tolerance or the active suppression of inflammatory responses to food and other orally ingested antigens is significantly ablated [37,38,39]. Pivotal relevance is assigned to intestinal macrophages that develop a unique phenotype called “inflammation anergy”, defined by a lack of an inflammatory response when encountering immunostimulatory microbes in the gastrointestinal tract [40]. Plasmacytoid and monocyte-derived dendritic cells (DCs) respond effectively to the microbiota and promote tolerogenic responses that result in anti-inflammatory mechanisms of immunoregulation [41,42]. More recently, a regulatory role for intraepithelial lymphocytes (IELs) responding to intestinal microbes has been described [43]. Microbe-derived ATP in concert with special lamina propria cells has been shown to activate the differentiation of CD4^+^ T helper cells to T helper 17 (Th17) cells, which function as inflammatory cell subsets essential for the recruitment of neutrophils to the site of infection [44]. The induction of Th17 cells at the intestinal level is a necessary mechanism of interphase between the adaptive and innate immune systems pivotal for the protection against pathogens. Certain bacteria, such as segmented filamentous bacterium (SFB), promote T cell differentiation into a Th17 phenotype in the small intestine [45,46,47]. Bacterial components, such as fragylysin, are proinflammatory factors [48,49]. Other bacterial components, such as lipid A, have been linked to neuroinflammatory processes and are associated with CNS diseases [50,51,52], including MS [53,54,55]. In consequence, the immune system requires counterbalance responses in order to control the potency of the necessary inflammatory processes triggered in response to infectious agents.

It is now widely accepted that intestinal bacteria are key regulators of the immune system by promoting peripherally induced regulatory T cells (Tregs). Tregs cells are characterized by the expression of the transcription factor forkhead box P3 (Foxp3) and promote an immunosuppressive microenvironment through cell-to-cell contact-dependent and -independent mechanisms. Tregs produce anti-inflammatory cytokines (IL-10 and/or transforming growth factor-beta, TGF-β) and regulate the proliferation of inflammatory cells. Certain clusters of Clostridia [56], polysaccharide A (PSA)-producing *B. fragilis* [57], Lactobacilli [58], or *Prevotella histolytica* [59] are examples of intestinal bacteria that promote the expansion of Tregs, and, as it will be summarized later, promote neuroprotection in experimental models of disease [59,60,61,62].

Fermenting symbiotic bacteria, including members of the family Lachnospiraceae [63], certain clusters of Clostridia [64], Bacteroides or Bifidobacteria [65], and others [66], digest complex polysaccharides, such as glycans, from fiber that are not digestible by human enzymes, producing short-chain fatty acids (SCFAs) as metabolites important for immunomodulation [67]. Acetate, propionate, and butyrate are the three most common SCFAs produced. SCFAs increase the amount of regulatory T cells by acting on the regulatory T cells through the GPCR43 receptor [64], decreasing inflammation. Together, microbial metabolites, such as SCFA, or microbial products, such as PSA, modulate the immune system’s anti-inflammatory pathways and function synergistically to decrease inflammation.

### 2.3. Impact of the Intestinal Microbiome beyond the Immune System

Studies done in GF animals support the hypothesis that the colonization of the intestine by microbes affect neurodevelopmental processes and behavior [68]. The presence or absence of intestinal microbiota regulate stress and anxiety responses and control processes of learning and memory [69]. Remarkably, GF mice receiving fecal content transplanted from severely depressed human patients acquire a phenotype with a behavior indicative of depression [70].

Different routes of interaction between the intestine and the brain could be responsible for the bidirectional effects observed in what it is now known as the microbiota–gut–brain axis [71]. The enteric nervous system (ENS) is known as the second brain. The ENS regulates the autonomous functions of the gastrointestinal tract, including peristaltic movements, enzyme synthesis, and neurotransmitter production, and serves as a modulator of the intestinal microbiota through the control of nutrient flow and oxygen availability [71]. The ENS communicates with the CNS through the parasympathetic and sympathetic nervous systems. The vagus nerve (VN) is the main nerve of the parasympathetic branch of the autonomic nervous system. The VN regulates metabolic homeostasis and is instrumental in immune system regulation [72]. The VN also regulates inflammatory processes through the hypothalamus–pituitary–adrenal (HPA) axis, the activation of the splenic sympathetic anti-inflammatory pathway, and the cholinergic anti-inflammatory pathway (CAP) (for a review, see [73]). The interaction between the VN and intestinal immune system is reciprocal. The stimulation of the VN alters cytokine levels [74,75,76], while inflammatory mediators, such as tumor necrosis factor alpha (TNF-α [77] and IL-1β [78], produced by innate immune cells in response to the microbiota activate the VN. The inflammatory activation of the VN triggers counterbalanced responses by the activation of CAP, which results in the control of inflammation by reduced cytokine production [79]. The selective activation of the efferent (brain to body) VN has been shown to suppress obesity-associated inflammation [80]. The HPA-dependent anti-inflammatory effects of acetylcholine produced as a result of VN activation results in reduced production of TNF-α IL-1β, IL-6, and IL-18 by macrophages, while the levels of anti-inflammatory IL-10 remain unchanged [81]. 

CNS disorders are often associated with gastrointestinal tract dysfunction, including constipation, inflammatory bowel disease, and inflammatory irritable syndrome [82]. Another possible consequence of an altered microbiota–gut–brain axis is intestinal barrier disruption, also known as leaky gut syndrome, a condition currently under extensive debate by the scientific community due to the potential involvement in autoimmune and inflammatory diseases [83]. Inflammation and the induction of disease increases intestinal permeability and exacerbates the proinflammatory responses observed at the intestinal level. The increase in the intestinal permeability observed in GF rats was reversed with probiotic treatments [84], and fecal transplantation restored the effects on the intestinal barrier induced by burn injury in mice [85]. The factors associated with leaky gut syndrome remain to be characterized. Nevertheless, it is now understood that intestinal dysbiosis results in intestinal epithelium disruption [86]. TNF-α is produced in response to intestinal microbes and is a key regulator of intestinal barrier permeability by controlling the expression of tight junction proteins [87]. In homeostatic situations, intestinal microbes may play a significant role in preventing leaky gut syndrome through the regulation of inflammatory processes occurring systemically and in other anatomical locations, such as the CNS. For instance, Treg induction by PSA-producing *B. fragilis* downregulates the production of TNF-α by human neutrophils in vitro [88]. The anti-TNF-α effects of PSA produced by *B. fragilis* are protective against experimental colitis [89], asthma [90], and EAE [60,61], and could be responsible for the prevention of leaky gut and increased inflammation at the intestinal level. Furthermore, SCFA produced as a result of bacterial fermentation of complex polysaccharides provides beneficial effects on intestinal barrier integrity [91]. The control of inflammation at the intestinal level can ultimately result in reduced neuroinflammation [92]. The intraperitoneal injection of LPS, as a surrogate for leaky gut and the presence of endotoxin outside the intestinal lumen, induces a systemic increase of TNF-α levels, including in the liver and brain. Even 10 months after injection the levels of TNF-α in the brain were still elevated in comparison to control mice [93]. Due to the association between the microbiota and leaky gut and the mechanistic effects of TNF-α in controlling intestinal barrier integrity, dysbiosis or treatment with intestinal microbes capable of controlling the levels of TNF-α could result in the exacerbation or reduction of neuroinflammation. As it is discussed in the next section, increased levels of TNF-α are observed in the brains of EAE mice [94] and MS patients [95].

The beneficial effects of the intestinal microbes and metabolites on the integrity of the epithelia by modulating the expression of tight junctions have also been observed at the blood–brain barrier (BBB), and are disrupted during EAE and MS. GF mice exhibit a disrupted BBB, including reduced epithelial expression of occludin and claudin and tight junction formation, whereas SCFA produced by intestinal fermenting bacteria can restore the integrity of the BBB in GF mice [96]. The disruption of the BBB could allow peripheral inflammatory cells to cross into the CNS and increase the potential for neurological inflammation [96]. Among SCFA, butyrate is critical in maintaining the integrity of epithelial barriers [97]. SCFA produced in the intestine could have profound effects on the parenchyma of the CNS in patients. In the brain, the maturation of the microglia is affected by SCFA [98]. In addition, there are other mechanisms by which SCFA could protect against neuroinflammation, including the differentiation of Tregs [64,99,100].

In addition to SCFA, other bacterial metabolites could regulate the extent of CNS inflammation and function. Many neurotransmitters and neuromodulators are expressed at the intestinal level. Serotonin levels, as well as tryptophan, are reduced in GF mice [101]. Intestinal spore-forming bacteria are critical in the production of serotonin [102]. Gamma amino butyric acid (GABA) is the principal inhibitory neurotransmitter in the CNS [103]. Certain intestinal lactic acid bacteria (LAB) of the *Lactococcus, Lactobacillus*, and *Streptococcus* genera produce varying levels of GABA by decarboxylating glutamate, through the enzyme glutamic acid decarboxylase (GAD) [104]. Many of these bacteria are naturally found in fermented dairy or other foods and GABA has been found to accumulate in some of these foods [104,105]. Both specific LABs and combinations of them have been investigated for their ability to alter GABA levels in treated mice. Mice who received a daily oral dose of *Lactobacillus rhamnosus* (JB-1) for 4 weeks had an increase in brain GABA levels [106], showing that GABA produced in the intestine can impact the brain. Furthermore, daily oral treatment for 28 days with the GABA producer *L. rhamnosus* (JB-1) resulted in increased GABA production in the mice’s brain, and when the VN was severed this effect was lost [107]. Other neurotransmitters produced in the intestine are norepinephrine by *Escherichia*, *Bacillus*, and *Saccharomyces* spp.; dopamine by *Bacillus* spp.; and acetylcholine by *Lactobacillus* spp. [108]. Additionally, the SCFAs produced by intestinal bacteria can cause the release of serotonin [109]. In patients suffering from MS, the levels of GABA are reduced [110,111], while the increase of GABAergic activity in EAE mice reduced the severity of the disease [112]. Antigen presenting cells expressing GABA A receptors (GABA_A_R) responded to the GABAergic compounds topiramate and vigabatrin in vitro by reducing the production of inflammatory cytokines IL-1β and IL-6 [112]. In this same study, the effects of GABAergic compounds did not affect the production of inflammatory cytokines by splenic T cells. However, another study showed that encephalitogenic T cells express GABA receptors and respond to physiological levels of GABA by reducing their proliferation and the production of cytokines, which may constitute a protective mechanism against neuroinflammation [113]. Thus, the interface between the immune system, neuroendocrine system, and the intestinal microbiota may constitute a novel avenue for the treatment of neuroinflammatory diseases.

## 3. The Intestinal Microbiome and MS

### 3.1. Experimental Evidence That Associates the Microbiome with MS

One of the most common animal models for MS is experimental autoimmune encephalomyelitis (EAE). This model has been used to study MS for over 100 years, and its origins are linked to the encephalomyelitis originally observed in Louis Pasteur’s rabies vaccine in the 19th century [114]. In this model, the animal is injected with antigens to the CNS as well as toxins to induce an immune response, which causes pathology as the immune system begins to attack the CNS [115]. Active EAE based on the immunization with self-antigens emulsified in complete or incomplete adjuvants and in the mouse models the use of pertussis toxin, and passive EAE dependent on the adoptive transfer of autoreactive T cells are both widely used in academic and pharmaceutical laboratories [116]. Although EAE should not be considered a surrogate for all complex aspects of MS disease in humans, some features of pathogenesis, including demyelination, neuroinflammation, axonal damage and loss, and symptoms, such as paralysis, closely resemble those of MS [114]. Nevertheless, EAE experiments have provided sufficient preclinical evidence for the development of currently approved MS drugs [116]. Moreover, experiments done using the EAE model were fundamental in establishing a link between the intestinal microbiota and CNS inflammatory diseases [117].

Initial studies done in EAE mice have shown that oral treatment with broad-spectrum antibiotics reduces and alters their intestinal microbiota, rendering animals more resistant in the development of EAE compared to mice treated with antibiotics intraperitoneally or untreated controls [118,119]. In both studies, the EAE severity was reduced in mice treated with antibiotics when compared with control mice, and the mechanism by which neuroinflammation was reduced was dependent on Tregs [118] or invariant natural killer (NK) cells [119]. The impact of the microbiota on the severity of EAE was later confirmed using GF mice. GF mice, born and raised under a complete absence of microbes or microbial products, show reduced severity of EAE when they are compared with mice that are raised under conventional experimental conditions, in the presence of environmental microbes [120,121]. The effects were shown to be based on reduced peripheral inflammatory responses. The monocolonization of GF mice with *SFB*, a known Th17 cell-inducing bacterium, restored the susceptibility to the disease through a mechanism dependent on the induction of Th17 cells at the intestinal level [120]. EAE mice treated with antibiotics and reconstituted PSA-producing *B. fragilis* were protected against the disease in a PSA-dependent mechanism, since their reconstitution of EAE with PSA-deficient *B. fragilis* did not confer protection against the disease [122]. Later studies showed that oral treatment with a purified form of PSA induced protection against EAE dependent on the induction of IL-10-producing regulatory T cell subsets, with a Foxp3-positive and -negative phenotype [60,61]. The induction of IL-10 production by the immune cells appears to be a pivotal mechanism of protection induced by intestinal microbes against neuroinflammatory EAE. Anti-inflammatory IL-10 is produced by monocytes, B cells, and activated T cells, including Tregs (Foxp3 positive) and Tr-1 cells (IL-10-producing activated CD4^+^ T cells that do not express Foxp3). During CNS inflammatory demyelination, IL-10 produced in response to intestinal bacteria might constitute a mechanism of protection. Recent studies of fecal microbiota transplantation (FMT) in GF mice appear to support the hypothesis that IL-10 produced in response to changes on the microbiota might protect against disease. Two independent studies showed that when the fecal content of MS patients is transferred to GF mice with reduced EAE susceptibility, the disease is exacerbated by a mechanism associated with an impaired function of IL-10-producing T cells [123,124]. Intestinal bacteria capable of inducing IL-10 production have also been shown to promote protection against EAE, such as combined formulations of *Lactobacilli* [62] or *Prevotella histolytica* [59]. Interestingly, the relative abundance of *P. histolytica* was found to be reduced in stool samples collected from MS patients when compared to healthy donors [59]. The oral treatment of EAE mice with the bacterium protected against disease by inducing IL-10-producing Tregs that were able to reduce the proinflammatory responses that characterize neuroinflammation in the model [59]. A recent study has shown that treatment with *P. histicola* is as effective as Copaxone^®^, an approved drug indicated for the treatment of MS for over two decades [125]. While Tregs play a protective role against experimental EAE, they also appear to be dysfunctional in those with MS, with reduced suppressive potency [126] and reduced frequencies in circulation [127]. Therapeutic approaches that target the intestinal immune responses that trigger enhanced frequencies and function of IL-10-producing regulatory T cells could constitute an effective and safer alternative to immunosuppressive drugs, many of which are associated with unwanted side effects [128].

Studies done in other models of CNS inflammatory demyelination have also provided evidence that the intestinal microbiota is an essential regulator of neuroinflammation. In a study by Mestre et al. [129], antibiotic treatment decreased the severity of neuroinflammation in the MS mice model Theiler’s murine encephalomyelitis virus (TMEV). In the marmoset model, a non-human primate model of MS, changes in the microbiota induced by dietary interventions also resulted in protection against the disease [130]. In conclusion, the evidence gathered in murine and non-human primate models of EAE suggest that the presence, alterations, and interventions of the intestinal microbiota modify the susceptibility of animals to CNS inflammatory demyelination or its severity. Moreover, EAE studies done in non-obese diabetic (NOD) mice that suffer from a biphasic form of the disease showed that the induction of neuroinflammation triggered significant changes in the intestinal microbiota [131]. These findings suggest that the interactions of the microbiota–gut–brain axis in the context of CNS are bidirectional and that undetermined factors triggered during disease are sufficient to promote dysbiosis.

### 3.2. Clinical and Epidemiological Evidence That Associate the Microbiome with MS

The etiological mechanisms of MS are not well understood. Results from experiments with the EAE model suggests that intestinal microbiota is an associated risk factor for disease [132]. Clinical studies of 16S ribosomal DNA of stool samples obtained from MS patients and healthy donors show differences in intestinal microbiota composition between cohorts. Interestingly, while the overall diversity of the microbiota is maintained in MS patients, statistically significant changes in the relative abundance of specific taxa are observed. Potential mechanisms regulating the microbiota–gut–brain axis were evaluated in FMT studies with stool isolated from MS patients and transplanted into GF mice [123,124]. Berer et. al. [118] found that microbiota from an MS-suffering monozygotic twin, when transplanted into the GF transgenic mouse model of spontaneous brain autoimmunity, resulted in a significantly higher incidence of autoimmunity compared to microbiota from the healthy twin. Similar results were obtained with microbiotas from three MS patients transferred into GF C57BL/6 EAE mice [124]. In both studies, amplicon sequencing analysis of 16S rDNA showed significant increases in the relative abundance of *Akkermansia muciniphila* and *Acinetobacter calcoaceticus* from MS patients, which are known to induce proinflammatory responses in human peripheral blood mononuclear cells [124]. In contrast, the relative abundance of *Parabacteroides distasonis*, known to stimulate anti-inflammatory Tregs in mice, were reduced in MS patients [124]. These studies support FMT as a potential therapeutic approach in the treatment of neuroinflammation. Indeed, clinical case reports of MS patients receiving FMT from healthy donors have been published [133]. In a single patient study, FMT from a healthy microbiota donor given to an MS patient resulted in disease stability for 10 years following transfer [133].

While the causative association between disease and microbiota remains to be elucidated, research suggests a clear association with disease and changes in the relative abundance of specific bacterial and archaea species [130]. A study published by Miyake et al. from 20 MS patients and 40 healthy individuals showed significant differences in the relative abundance of 21 microbial species in MS patients’ samples. Specifically, species belonging to *Clostridia* clusters XIVa and IV, and Bacteroidetes were decreased [134]. Although protective effects of PSA-producing *B. fragilis* have been documented and previously described, the protection appeared to be PSA dependent [122]. Moreover, Bacteroidetes and more specifically the genus *Bacteroides* comprise other species associated with human disease, including colorectal cancer [135]. Other studies have shown a general increase in the relative abundance of Clostridia [13,124]. This may be due to the fact that the *Clostridia* genera are highly diverse [136] and contain species known to induce both anti-inflammatory [56] and inflammatory responses [137]. Beneficial or detrimental effects due to increased or decreased relative abundance of Clostridia may be dependent on the high diversity of species within these genera. In other studies, fecal microbiome analyses of 31 MS patients and 36 age- and gender-matched controls showed an increase in *Pseudomonas*, *Mycoplana*, *Haemophilus*, *Blautia*, and *Dorea* genera in MS patients; the control group had an increase of the abundance of *Parabacteroides*, *Adlercreutzia*, and *Prevotella* genera [138]. In another study with 60 MS patients and 43 healthy controls, increases in the abundance of *Methanobrevibacter* and *Akkermansia*, as well as decreases in *Butyricimonas* genera in MS patients were found [139]. As discussed in the previous section, a significant reduction in the relative abundance of *P. histicola* was observed in stool samples of MS patients when compared to healthy donors. Reconstitution of EAE mice with the bacterium protected against the disease in an IL-10-producing Treg-dependent mechanism [59,125]. Some of these changes in bacterial composition and levels may also be in response to the disease, as it is known that MS progression increases fecal transit time [140]. Increased fecal transit time has been shown to lead to changes in the intestinal microbiome of specific taxa, such as the archaeal methane producer *Methanobrevibacter* [141].

Ethnic variability appears to also be a contributing factor to microbiome variability and disease severity. In a study looking at the differences in the microbiome of MS patients between different ethnicities, the microbiota composition of Caucasian, Hispanic, and African American individuals with MS varied compared to ethnicity-matched healthy controls [13]. In addition, differences in the disease patterns and severity have also been observed between ethnicities [14]. In general, Hispanics and African Americans suffer from a more severe form of MS than Caucasians [142], while the intestinal microbiota might also differ among ethnic groups regardless of geographical location [15]. A significant increase was observed in the relative abundance of the genus *Clostridium* in the stool samples from MS patients of the three ethnicities tested when compared to ethnicity-matched controls. However, larger data sets are needed in order to more definitively establish a link between microbiota differences among ethnic groups and their disease profiles.

Disease-modifying treatments (DMTs) also are known to impact the intestinal microbiome composition [143]. The microbiota of patients undergoing disease-modifying therapies had an increase of *Prevotella* and *Sutterella* genera and a decrease of *Sarcina* genera compared to untreated patients [139]. Many common MS treatments have been documented to inhibit the growth of certain microbes, such as *Clostridium perfringens* [144]. Interestingly, the relative abundance of *C. perfringens* in the intestinal microbiota of patients suffering from neuromyelitis optica (NMO) is increased when compared to healthy individuals [145]. NMO is another autoimmune disease of the CNS, closely related to MS, and characterized by the autoreactivity of Th17 cells against aquaporin-4. More recently, the presence of epsilon toxin from *C. perfringens* was found in a significantly higher percentage of MS patients than in healthy controls in a study including MS patients, clinical isolated syndrome (CIS) patients, and optic neuritis patients [146]. However, the functional relevance of the changes that DMT promote in the intestinal microbiota remain to be elucidated. Whether these effects are the direct result of DMT drugs on microbes or indirect interactions with immune cells and soluble factors released by immune cells in response to treatment needs further analysis.

## 4. Interventions of the Intestinal Microbiota as a Treatment of MS

There are many MS therapies currently approved by the US Food and Drug Administration (FDA) (Table 1). Most treatments focus on suppressing the immune system, which can lead to increased risk of infection. MS therapies are not universally effective, and responses among individuals can be highly variable.

### 4.1. Experimental Interventions of the Microbiota

#### 4.1.1. Antibiotics

Antibiotics may be used to kill off inflammatory microbes. The oral MS therapeutics Fingolimod, Teriflunomide, and dimethyl fumarate (DMF) are known to inhibit *C. perfringens* growth [144]. While mice treated with antibiotics and GF mice are more resistant to developing EAE [119], a healthy microbiome is important in the protection against inflammation. Antibiotics may further be explored as a therapeutic option, but this will require the identification and characterization of antibiotics targeting only inflammation-inducing bacterial species.

#### 4.1.2. Bacteriophage Therapy

Bacteriophage may provide the specificity required to manipulate the microbiota. Bacteriophages (phages) are viruses that target and infect specific species of bacteria and therefore may be a useful approach to reduce or eliminate inflammatory bacterial species from an MS patient [147]. Specific targeting of inflammatory bacterial will require a deeper understanding as to which specific species are contributing to inflammation and disease. One of the disadvantages of this approach is the limitations of our knowledge as to the complex interspecies interactions present in the intestinal microbiota. Disrupting microbial interaction networks that promote the generation of beneficial metabolites could result in deleterious consequences to the host immune system [148].

#### 4.1.3. Fecal Microbiota Transplantation (FMT)

FMT from a healthy donor has been explored as a potential treatment aimed to re-establish a healthy microbiota in the recipient. This approach could be considered microbiota agnostic, as the entire network of microbial species are simultaneously transplanted from a healthy donor. While the health of the donor must be established prior to transplantation, there remains the risk of transplanting microbes that may not be well tolerated by the recipient. As discussed in the previous section, FMT from a healthy donor to an MS patient induced compositional changes in the recipient’s microbiota and reduced the severity of neurological symptoms [133]. However, larger studies are needed to determine the feasibility and efficacy of FMT as a therapeutic approach to treat MS.

#### 4.1.4. Microbial-Derived Products

The purification of microbial products capable of inducing immunomodulation could be a safe mechanism to induce protection or as a therapeutic intervention for the treatment of MS. EAE studies done with purified PSA produced by *B. fragilis* suggest that oral treatment with microbial products could be beneficial in the protection against neuroinflammation without the potential challenges of autoreactive antigens [60,61]. Furthermore, PSA modulates the phenotype of human T cell subsets, inducing anti-inflammatory phenotypes dependent on IL-10 production in healthy donors [88] and MS patients [149].

#### 4.1.5. Probiotics

Probiotics are also being explored for the treatment of neuroinflammation. Probiotics are commercially available formulations of living organisms that when consumed orally have been shown to confer numerous health benefits [147]. Probiotics are most commonly LABs belonging to the *Lactobacillus* or *Lactococcus* genera or *Bifidobacterium* and are generally recognized as safe (GRAS) by the FDA [147]. Studies using single species/strains have also shown promising results against neuroinflammation. Oral treatment with *Lactobacillus reuteri* protects against murine EAE by modulating the intestinal microbiota [150]. During EAE, the relative abundance of genera considered beneficial, such as *Bifidobacterium*, *Prevotella*, and *Lactobacillus*, were reduced when compared to healthy controls. Treatment with *L. reuteri* reduced Th1 and Th17 proinflammatory responses in mice, restored healthy intestinal microbiota, and induced protection against disease [150]. *Lactobacillus helveticus* SBT2171 also protects against EAE in mice by reducing the differentiation of Th17 cells [151]. The protective effects of probiotic mixes have also been tested in EAE models. Individual treatment with Lactobacillus strains *Lactobacillus paracasei* DSM 13434, *Lactobacillus plantarum* DSM 15312, or *Lactobacillus plantarum* DSM 15313 is associated with decreased levels of myelin oligodendrocyte glycoprotein (MOG)-reactive T cells and reduced EAE severity but did not suppress established EAE symptoms [62]. However, treatment with a mixture containing all three strains successfully reversed established EAE and was associated with systemic IL-10 release and induction of Tregs in lymph nodes, periphery, and CNS [62]. A more recent study compared the protective effects of two commercially available probiotic mixes against EAE induced in C57BL/6 mice: Lactibiane Iki, composed of *Bifidobacterium lactis* LA 304, *Lactobacillus acidophilus* LA 201, and *Lactobacillus salivarius* LA 302; and Vivomix, composed of Lactobacilli, Bifidobacteria, and *Streptococcus thermophilus* [152]. Both probiotics reduced the extent of demyelination and T cell levels in the spinal cords of EAE mice, modified the intestinal microbiota, and affected the levels of antigen presenting cell (APC) immune cells. Lactibiane Iki treatment additionally resulted in increased expression of programmed death-ligand 1 (PD-L1) and reduced expression of major histocompatibility complex (MHC) class II and CD80 in APCs, suggesting that this probiotic treatment promoted an immune-suppressive tolerogenic phenotype in APCs. Similarly, treatment with Lactibiane Iki increased the populations of Tregs in the periphery while reducing the percentages of plasma cells in circulation. Lactibiane Iki furthermore was able to reduce the severity of EAE [152]. As we continue to learn how probiotics intersect with the immune system, targeted probiotic treatment will allow us to modify the intestinal microbiota to decrease inflammation, promote immune health, and to prevent or ameliorate symptoms associated with MS and other inflammatory diseases.

### 4.2. Genetic Design of Probiotics with Enhanced Protective Phenotype?

Scientists and clinicians alike have a keen interest in designing probiotics to enhance and expand their therapeutic potential. Engineered probiotics have been developed as potential therapeutics for a variety of inflammation-associated and human diseases, including MS, irritable bowel disease (IBD), and diabetes [153,154,155]. The anti-inflammatory cytokine IL-10 has a central role in downregulating inflammatory cascades [156] and is a target for engineered probiotics. A *Lactococcus lactis* strain expressing murine IL-10 is both preventative and therapeutic in the dextran sodium sulfate (DSS)-induced mouse colitis model [154]. Expression of IL-10 in combination with the type 1 diabetes autoantigen glutamic acid decarboxylase (GAD65) by *L. lactis* reversed symptoms in the non-obese diabetic (NOD) mouse model [157]. Probiotics have also been engineered to address injury promoting inflammation due to reactive oxygen species (ROS); the induction of expression of the oxidative stress protective enzyme superoxide dismutase by various Lactococci and Lactobacilli strains is protective in both mice (DSS induced and IL-10 deficient) and rat (TNBS) colitis models [155,158,159]. *Lactococcus lactis* expressing IL-35 protects against rheumatoid arthritis (RA) in mice [160]. Human angiotensin converting enzyme 2 (ACE2), which is associated with reduced inflammation and oxidative stress in humans, when expressed in *Lactobacillus casei* effectively reduced retinopathy symptoms in two diabetic retinopathy mouse models [161]. Heat shock proteins (*Mycobacterium* Hsp65) that act as immunomodulatory factors through their role as autoantigens and promotion of Treg survival, when expressed in *L. lactis*, are protective against both EAE and DSS-induced colitis [162,163]. Enzymes detected in low levels or those that are absent in disease states have also been targeted through engineered probiotics. Colonic tissue from IBD patients shows decreased levels of Elafin, an endopeptidase that prevents elastase-mediated tissue proteolysis associated with IBD [162]. *Lactococcus lactis* and*L. casei* expressing human Elafin decrease inflammation in both acute and chronic IBD mouse models [164]. Engineered probiotics are proving an effective delivery method for immunomodulatory molecules that prevent or relieve symptoms in additional human inflammatory animal disease models, including EAE, diabetes, colitis, and arthritis [157,162]. The results of these studies support the rationale development of novel probiotics for the treatment of MS. The pivotal importance of immunomodulatory IL-10, IL-35, Tregs, and metabolites, such as SCFA and neurotransmitters, and other protective factors identified in these animal models and clinical studies will provide insight into the development of novel therapeutic targets for use in genetically engineered probiotics [165]. Figure 1 depicts the mechanisms of action proposed for probiotics that could result in the control of neuroinflammation-mediated demyelination.

#### 4.2.1. Probiotic Strain Choice for Engineering?

Lactococci and Lactobacilli are particularly well-suited, and therefore most commonly used, for the development of engineered probiotics. In addition to the GRAS designation that designates them safe for human consumption (FDA), Lactococci and Lactobacilli are Gram-positive bacteria lacking endotoxin (lipid A) and with a single membrane for protein display. In addition, there are a suite of molecular tools available for their genetic manipulation [166,167,168,169]. In the studies presented above, *Lactobacillus* strains [161,170] or *L. lactis* [154,157,162,163], and in some cases both [153,155], were engineered for use in these therapeutic studies. In the experimental colitis and EAE studies testing both a *Lactobacillus* strain and *L. lactis* for recombinant protein delivery, no significant difference in protection conferred against experimental colitis and EAE, respectively, was observed [155,164]. One advantage of using Lactobacilli over Lactococci as a therapeutic probiotic is their ability to colonize the human gastrointestinal (GI) tract [171]; Lactococci survive transit through the human GI but are unable to colonize it [172,173]. However the molecular tools available for *Lactococcus* spp., *Lactococcus lactis* in particular, far exceed what is available for Lactobacilli and *L. lactis* remains the primary model for probiotic engineering [174,175]. Furthermore, if an *L. lactis*-based probiotic treatment would be no longer needed, the lack of colonization would facilitate the clearance of the therapeutic from the GI tract of the patient.

#### 4.2.2. Tools for Probiotic Engineering

Multiple plasmid systems have been developed to allow straightforward gene introduction and expression in *L. lactis*. Key differences in these systems include variations in the levels of expression, regulated or constitutive expression, and target protein destination.

Expression of heterologous proteins is primarily controlled at the level of transcription initiation by promoters that confer constitutive expression at a fixed level or in response to an exogenously added inducer or environmental conditions. An advantage in using a regulated expression system is to allow temporal control of protein expression or the ability to express an otherwise toxic gene product. The most widely used inducible expression system is the NIsin Controlled Expression (NICE) system, in which the P*_nisA_* promoter is induced by the NisA peptide (nisin) through the two-component system NikS and NikR. This system can achieve a dynamic range of expression (up to 1000 fold) by adding increasing concentrations of nisin [176,177]. Additional inducible expression systems include the agmatine controlled expression (ACE) [178,179], Zirex [180,181], P_(Zn)*zitR*_ [182], P*_xylT_* [183], stress inducible expression (SICE) [184], and P170 systems [185]. These systems are regulated by a variety of exogenous molecules (e.g., agmatine, Zn^2+^ and xylose) or changes in environmental conditions. The SICE system, for example, is a stress-inducible expression system, where P*_groESL_*-directed expression is induced by heat shock, low pH, or UV irradiation [184]. Inducible promoter systems offer numerous expression control options; however, achieving induction or a necessary environmental condition may prove challenging in the host. Promoters that confer constitutive expression of a target gene are particularly well suited to in situ purposes as they allow for the steady-state delivery of the desired product and there are no requirements for induction or cloning of additional regulatory elements. Both native [186,187] and optimized [188] *L. lactis* constitutive promoters are available that provide a range of expression over four orders of magnitude, from low (e.g., P32 and P44) to high (e.g., P21, P23, P59, P2, P3, P5, and P8) expression.

Depending on the therapeutic purpose, expressed heterologous proteins can be directed to different *L. lactis* compartments, the cytoplasm, the cell surface for display (membrane or cell wall) [189,190,191], or secreted to the extracellular environment [168]. No additional modifications are required for cytoplasmic localization of heterologous proteins, but N- and sometimes C-terminal signal peptides are required for localization to other locations. Heterologous proteins’ secretion requires fusion to *L. lactis* N-terminal signal peptides (SPs) that direct efficient secretion (Sec) pathway-dependent transport across the cell membrane. The mostly widely used SP is derived from the major secreted *L. lactis* protein LcoB/Usp45 [192], but several other native (SP_ExpE_ [193], SP_310_ [194], AC9, and BL1 [195]) and engineered (SP2_310mut2_ [196] and SP_LcoP_ [197]) *L. lactis* SPs are available. Secretion of and stability of heterologous proteins can also be enhanced by targeting SP-dependent secretion Sec machinery [198,199], *L. lactis* proteases [200,201,202], and *ybdD* [203]. For example, overexpression of the *L. lactis* SP signal peptidase, SipL [204], or mutation of the cytoplasmic (ClpP, [200,201]) and major secreted (HrtA, [202]) proteases, or *ybdD* [203], also increase overall levels of secreted proteins.

*Lactococcus lactis* surface display of heterologous proteins can be targeted to the cell membrane or cell wall. For membrane targeting, proteins can be fused to transmembrane domains but are more commonly fused to an N-terminal lipobox-containing lipoprotein anchor, such as lipoprotein BmpA from *L. lactis* [205]. Cell wall display of heterologous proteins requires both the Sec-dependent N-terminal SP for transport across the cell membrane and a C-terminal signal peptide (CSP). Fusion to the sortase system-dependent CSP (LPXTG) [206,207] or the *L. lactis* AcmA autolysin CSP (LysM motif) [208] direct protein attachment to the cell wall. There are many options for combination of the expression features discussed above, to achieve appropriate localization and therapeutic concentrations of heterologous proteins [209]. For the most widely used NICE systems, there are engineered *L. lactis* strains and plasmids commercially available that combine some of the above discussed features [175,177,210]. The plasmid copy number of exogenously replicating plasmids containing the target gene (s) to be expressed can be variable between cells and often requires selective pressure for maintenance, e.g., by culture with antibiotics or in the absence of a carbon source [211,212]. Similarly, plasmid copy number variability could alter the therapeutic concentrations of the desired target and plasmid selection in situ when considering engineered probiotics for clinical purposes may be challenging (e.g., additional requirement for a selective agent or diet). Incorporating engineered constructs through integration into the *L. lactis* chromosome overcomes these challenges; there is no longer a requirement for selection and the target gene copy number remains constant [213,214]. Plasmids that allow the integration of engineered constructs into the *L. lactis* chromosome have been developed based on site-specific or homologous recombination and selection for a plasmid marker (e.g., antibiotic resistance). These plasmids are either non-replicative in *L. lactis* (e.g., ColE1 derivatives) or are conditionally replicative (e.g., thermosensitive [215] and *repA* deficient [216]). Incorporation of bacteriophage features within the plasmid, such as *attP* and integrase, has been used to facilitate site-specific recombination at corresponding *attB* chromosomal loci [217,218]. Homologous recombination requires a small region of the *L. lactis* chromosomal DNA on the plasmid and usually occurs by Campbell-like integration [213]. *Lactococcus lactis* chromosomal loci used for homologous recombination-based integration include *leuA*, *tel*, and *llmg_pseudo_10*a [219,220]. The resulting cointegrate between the plasmid and chromosome can be unstable and leaves plasmid DNA, including the selectable marker (e.g., antibiotic resistance gene), in the chromosome. To address this, integrative plasmids have been designed with a variety of counterselection or screening strategies that involve two single recombination events [221,222]. The first recombination event forms the cointegrate, as described above, and the second recombination event either leaves the unmarked engineered construct on the chromosome (no plasmid DNA) or restores the original chromosomal locus [219]. Plasmid-based copies of the *L. lactis* genes *upp*, *oroP*, or *pheSA312G* provide counterselection for *L. lactis* in which the second recombination event has occurred [221,223,224]; cells expressing high levels of *pheSA312G*, for example, are sensitive to the phenylalanine analog *p*-chloro-phenylalanine [224]. Integration strategies have been combined in a number of ways. For example, culture of the thermosensitive plasmid pG+-5P-*pheS** at 37 °C forces integration, and subsequent culturing at 28 °C allows plasmid excision; the genomes of *p*-chloro-phenylalanine-resistant strains are then analyzed by PCR for the gene replacement event [224]. While time consuming, these strategies generate an engineered probiotic with an unmarked stable copy of the engineered construct and no requirement for selection. 

Among technologies being implemented and developed to improve efficiency in engineering probiotics are RNA-guided Clustered Regularly Interspaced Short Palindromic Repeat (CRISPR)/Cas9 genome editing and recombineering [169,225,226,227]. The CRISPR/Cas9 system has been used as a mutagenesis screening strategy. Co-expression of Cas9 with the target short guide RNA has been used for counterselection against cells lacking the desired chromosomal modification [224,226]. An ssDNA recombineering method for gene mutation in *L. lactis* was recently described [169,227]. This method introduces an ssDNA oligonucleotide with the desired change and incorporates the change into the chromosome by double-crossover homologous recombination; this inefficient event is enhanced by the expression of an exogenous *Enterococcus faecalis* recombinase, RecT [169]. In combination with CRISPR/Cas9 counterselection, ssDNA recombineering can reduce the *L. lactis* construction time from several weeks to 72 h. [169]. Additional development is still needed for recombineering to replace more time-consuming homologous recombination strategies, as there are size limitations on the deletion (100 bp) or insertions (34 bp) conferred by this system [169].

## 5. Conclusions

The intestinal microbiota plays an important role in the training and development of a healthy immune system, broadly influences inflammatory immune processes, and has significant impact on the neuroinflammatory disease multiple sclerosis. As we continue to elucidate the role specific microbial species contribute to the progression of this disease, additional therapeutic opportunities will arise. With advances in genetic engineering, we now have the capability of designing smart probiotics to bring the fight against MS to the battlefield of the intestinal microbiome.

## Figures and Tables

**Figure 1 diseases-08-00033-f001:**
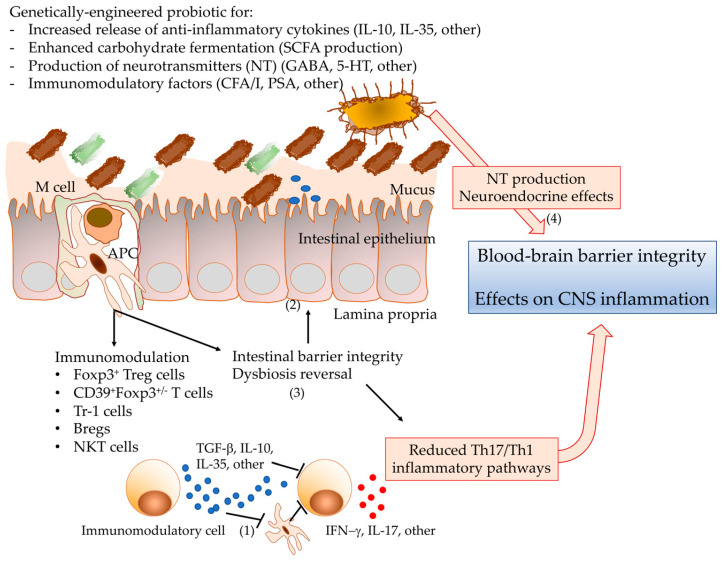
Potential mechanisms of action for neuroinflammation-targeting probiotics. Microbial factors, such as colonization factor antigen I (CFA/I) fimbriae or polysaccharide A (PSA), promote the differentiation of immunoregulatory cell subpopulations. Modifications of the balance between inflammatory and immunoregulatory cells by inhibitory effects on antigen presenting cells (APCs), or directly targeting proinflammatory cell populations with anti-inflammatory cytokines (1) suppress experimental inflammatory demyelination. Microbial metabolites, such as SCFA, can also regulate the integrity of the intestinal barrier as well as the blood–brain barrier (2). Probiotics have also been shown to restore intestinal homeostasis by balancing the microbiota (3). Furthermore, the design of probiotics specifically directed to increase the production of other metabolites, such as neurotransmitters, could affect the neurobiology of CNS inflammatory diseases (4). APC: antigen presenting cell; Breg: regulatory B cell; CFA/I: colonization factor antigen I of enteroxigenic Escherichia coli; CSFA: short-chain fatty acids; CNS: central nervous system; 5-HT: 5-hydroxytryptamine (5-HT); IL: interleukin; IFN: interferon; NKT: natural killer T cell; PSA: polysaccharide A; TGF: transforming growth factor.

**Table 1 diseases-08-00033-t001:** United States (US) Food & Drug Administration (FDA)-approved treatments for multiple sclerosis.

Injectable Medications			
Therapeutic	Target	Most Common Side Effects *	Proposed Mechanism of Action
Interferon beta-1a	Inflammation	Headache, flu-like symptoms, injection site pain, inflammation.	Anti-inflammatory effects
Interferon beta-1b	Inflammation	Flu-like symptoms, headache, injection site reactions, injection site skin breakdown, low white blood cell count.	
Glatiramer acetate	Inflammation	Injection site reactions, flushing, shortness of breath, rash, chest pain.	Immunomodulation
**Oral Treatments**			
Teriflunomide	Inflammation	Headache, hair thinning, diarrhea, nausea, abnormal liver tests.	Controls proliferation of auto-reactive cells.
Fingolimod	Inflammation	Headache, flu-like symptoms, diarrhea, back pain, abnormal liver tests, sinusitis, abdominal pain, pain in extremities, cough. It can slow heart down.	Sphingosine 1-phosphate receptor modulator: Blocks lymphocyte egress from lymph nodes.
Cladribine	Inflammation	Upper respiratory infection, headache, low white blood cell counts.	Immunosuppressive effects on lymphocytes (T and B cells).
Dimethyl fumarate	Inflammation	Flushing, gastrointestinal issues.	Immunomodulatory and antioxidative effects.
**Intravenous Infusion Treatments**			
Alemtuzumab	Inflammation	Rash, headache, fever, nasal congestion, nausea, urinary tract infection, fatigue, insomnia, upper respiratory tract infection, herpes viral infections, hives, itching, thyroid gland disorders, fungal infection, pain in joints, extremities and back, diarrhea, vomiting, flushing. Infusion reactions common.	Humanized anti-CD52 monoclonal antibody—depletes CD52 + lymphocytes.
Ocrelizumab	Inflammation	Infusion reactions; increased risk of infections; possible increase in malignancies, including breast cancer.	Humanized anti-CD20 monoclonal antibody: Targets CD20 + B cells
Natalizumab	Inflammation	Headache, fatigue, joint pain, chest discomfort, urinary tract infection, lower respiratory tract infection, gastroenteritis, vaginitis, depression, pain in extremity, abdominal discomfort, diarrhea, and rash. It increases Risk of progressive multifocal leukoencephalopathy (PML), a deadly opportunistic viral infection of the brain.	Anti-α4β1-integrin monoclonal antibody. Blocks T cell migration to CNS

* Source: National MS Society.

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
