# Peer review of "The Microbiome as a Therapeutic Target for Multiple Sclerosis: Can Genetically Engineered Probiotics Treat the Disease?"

_diseases, 2020, doi:10.3390/diseases8030033_

Round 1

Reviewer 1 Report

Authors Kohl et al. submitted their manuscript entitled „The microbiome as a therapeutic target for multiple sclerosis: Can genetically engineered probiotics treat the disease?“

Metazoan hosts have developed in the presence of complex microbiota that consists of bacteria, archaea, fungi, yeast, protozoa, and viruses. In the last decade, knowledge of relationships of host and its microbiota increased, and specific modulation of the microbiome became part of personalized medicine. The authors reviewed the relation of microbiota to various diseases and paid their primary attention to multiple sclerosis (MS).

I have several notices and recommendations:

L66-69: The definition of the microbiome is complicated and not clear. Please, simplify and clarify it.

Table 1 - I did not find an explanation of the abbreviation of the FDA. It is not possible to expect a priori that everybody knows this abbreviation.

L199: I think that occludin should be singular in contrast to that it exists in more forms.

L257-258: It is not clear if you wrote about ex GF (conventionalized) mice reared in conventional conditions or about conventional mice. Please, rephrase the sentence.

L417: The abbreviation FDA was used several times without its explanation. Please, explain it when it is used at first.

L468: Please, correct IDB to IBD.

L514-607: This part of the text seems to be inadequately detailed and specialized concerning the whole manuscript text. Please, consider a simplification of the text.

Author Response

We would like to thank the reviewers for all the positive remarks and for the notes provided. Please find below the specific responses to the points raised. We have modified the manuscript accordingly.

Reviewer 1:  Metazoan hosts have developed in the presence of complex microbiota that consists of bacteria, archaea, fungi, yeast, protozoa, and viruses. In the last decade, knowledge of relationships of host and its microbiota increased, and specific modulation of the microbiome became part of personalized medicine. The authors reviewed the relation of microbiota to various diseases and paid their primary attention to multiple sclerosis (MS).

Response: We appreciate your positive remarks and your constructive critique. Below are our responses to each of the concerns raised.

L66-69: The definition of the microbiome is complicated and not clear. Please, simplify and clarify it.

Response: Thank you for your suggestion. We have modified the section accordingly. In order to respond appropriately to the comment, we did a search on the term “microbiome” and identified a recent letter published in the journal “Microbiome” that highlights the need for standardization of the use of the term [1]. We believe it is relevant to include this reference to our manuscript, in order to clarify the concepts “microbiota” and “microbiome”.

Table 1 - I did not find an explanation of the abbreviation of the FDA. It is not possible to expect a priori that everybody knows this abbreviation.

Response: Thank you. We modified the text defining the abbreviation for (US Food & Drug Administration).

L199: I think that occludin should be singular in contrast to that it exists in more forms.

Response: We modified the text accordingly.

L257-258: It is not clear if you wrote about ex GF (conventionalized) mice reared in conventional conditions or about conventional mice. Please, rephrase the sentence.

Response: We apologize for any confusion. We have modified the text and clarified the statement:  germ-free mice, born and raised under the complete absence of microbes or microbial products show reduced severity of EAE when compared with mice that are raised under conventional experimental conditions, in the presence of environmental microbes. 

L417: The abbreviation FDA was used several times without its explanation. Please, explain it when it is used at first.

Response: We defined the acronym the first time used in the text. Thank you.

L468: Please, correct IDB to IBD.

Response: Thank you. We modified the text accordingly.

L514-607: This part of the text seems to be inadequately detailed and specialized concerning the whole manuscript text. Please, consider a simplification of the text.

Response: We appreciate the concern raised by the reviewer. In the revised manuscript we have simplified the section (new lines 540 - 558, 565 - 572, and 713 - 722).  Although the work is not intended to be a methods article and we agree with the reviewer that the information provided was detailed, we wanted to prepare a manuscript that can be informative for those readers interested in pursuing genetic modifications of probiotics. Accordingly, the new version of the manuscript shows a reduced form of the section but still includes what we believe is relevant for those potentially interested in genetically-modifying bacteria to alter the course of multiple sclerosis. 

Reviewer 2 Report

Review manuscript described by Hannah Kohl and colleagues “The microbiome as a therapeutic target for multiple sclerosis: Can genetically engineered probiotics treat the disease?” is very interesting and significant in relation to novel strategies to achieve the microbiome modulation. Authors did a good job to perform this review and included information from recent research articles. The manuscript is well organized and easy to read. Authors amaizingly summarizes the impact of the microbiota on the immune system and MS, they review possible treatments for microbiota modulation with its pros and cons to finally develop the object of the review, the design of genetically engineered probiotics. I personally like the way the authors have approached the issue and how they described the use of genetic engineering for the design of smart probiotics, certainly a novel field to explore in the near future. I've really enjoyed reading this job, so I just have few suggestions:

Figure 1 is quite blurry, there are text part that are hard to read

Line 468: IDB must be IBD¿?

Author Response

We would like to thank the reviewer for all the positive remarks and for the notes provided. Please find below the specific responses to the points raised. We have modified the manuscript accordingly.

Figure 1 is quite blurry, there are text part that are hard to read

Response: We apologize for uploading a low-quality image. We have increased the resolution of the figure accordingly. Thank you.

Line 468: IDB must be IBD¿?

Response: Thank you. We modified the text accordingly.
